# Gaussian Connectivity-Driven EEG Imaging for Deep Learning-Based Motor Imagery Classification

**DOI:** 10.3390/s26010227

**Published:** 2025-12-29

**Authors:** Alejandra Gomez-Rivera, Diego Fabian Collazos-Huertas, David Cárdenas-Peña, Andrés Marino Álvarez-Meza, German Castellanos-Dominguez

**Affiliations:** 1Signal Processing and Recognition Group, Universidad Nacional de Colombia, Manizales 170003, Colombia; dfcollazosh@unal.edu.co (D.F.C.-H.); amalvarezme@unal.edu.co (A.M.Á.-M.); cgcastellanosd@unal.edu.co (G.C.-D.); 2Automatics Research Group, Universidad Tecnológica de Pereira, Pereira 660003, Colombia; dcardenasp@utp.edu.co

**Keywords:** deep learning, EEG, motor imagery, gaussian connectivity, imaging, explainability

## Abstract

Electroencephalography (EEG)-based motor imagery (MI) brain–computer interfaces (BCIs) hold considerable potential for applications in neuro-rehabilitation and assistive technologies. Yet, their development remains constrained by challenges such as low spatial resolution, vulnerability to noise and artifacts, and pronounced inter-subject variability. Conventional approaches, including common spatial patterns (CSP) and convolutional neural networks (CNNs), often exhibit limited robustness, weak generalization, and reduced interpretability. To overcome these limitations, we introduce EEG-GCIRNet, a Gaussian connectivity-driven EEG imaging representation network coupled with a regularized LeNet architecture for MI classification. Our method integrates raw EEG signals with topographic maps derived from functional connectivity into a unified variational autoencoder framework. The network is trained with a multi-objective loss that jointly optimizes reconstruction fidelity, classification accuracy, and latent space regularization. The model’s interpretability is enhanced through its variational autoencoder design, allowing for qualitative validation of its learned representations. Experimental evaluations demonstrate that EEG-GCIRNet outperforms state-of-the-art methods, achieving the highest average accuracy (81.82%) and lowest variability (±10.15) in binary classification. Most notably, it effectively mitigates BCI illiteracy by completely eliminating the “Bad” performance group (<60% accuracy), yielding substantial gains of ∼22% for these challenging users. Furthermore, the framework demonstrates good scalability in complex 5-class scenarios, performing competitive classification accuracy (75.20% ± 4.63) with notable statistical superiority (*p* = 0.002) against advanced baselines. Extensive interpretability analyses, including analysis of the reconstructed connectivity maps, latent space visualizations, Grad-CAM++ and functional connectivity patterns, confirm that the model captures genuine neurophysiological mechanisms, correctly identifying integrated fronto-centro-parietal networks in high performers and compensatory midline circuits in mid-performers. These findings suggest that EEG-GCIRNet provides a robust and interpretable end-to-end framework for EEG-based BCIs, advancing the development of reliable neurotechnology for rehabilitation and assistive applications.

## 1. Introduction

Engineering has emerged as a cornerstone in addressing pressing global health challenges, as emphasized in UNESCO’s Engineering for Sustainable Development report (2021), which recognizes the discipline as a central pillar of the 2030 Agenda. Among the Sustainable Development Goals, SDG 3—“ensure healthy lives and promote well-being for all at all ages”—underscores the need for accessible and affordable technologies to strengthen medical diagnostics and healthcare delivery [1]. Within this framework, brain–computer interfaces (BCIs) have gained increasing international attention for their potential to revolutionize human–machine interaction in clinical, rehabilitative, and assistive domains. Beyond their scientific and societal impact, BCIs are also economically significant: the global market is projected to reach USD 2.21 billion by 2025 and expand further to USD 3.60 billion by 2030, with a compound annual growth rate (CAGR) of 10.29 [2]. This convergence of societal need, technological innovation, and market growth highlights BCIs as a key enabler for sustainable health solutions.

Within this landscape, electroencephalography (EEG) has become a foundational technology for BCI implementation, owing to its non-invasive nature, low cost, portability, and high temporal resolution [3]. EEG captures the brain’s electrical activity through scalp-mounted electrodes and enables the use of advanced signal processing techniques such as event-related potentials (ERPs), which are widely used to assess cognitive and motor functions [4]. These features have established EEG as the technological backbone of many modern BCI platforms, allowing for the decoding of neural signals to control external devices without requiring muscular input [5]. Among the various paradigms, motor imagery (MI)—the mental rehearsal of movement without physical execution—has demonstrated remarkable clinical potential in post-stroke rehabilitation, neuroprosthetic control, and assistive technologies such as robotic wheelchairs and virtual spellers [6].

Despite its versatility, EEG-based BCI systems face inherent structural limitations, primarily due to the physical and physiological nature of the recorded brain signals [7]. These constraints compromise both signal quality and interpretability, directly affecting their clinical and functional applicability [8]. One of the most critical challenges is inter-subject variability, which introduces pronounced inconsistency in neural activation patterns and undermines the generalization of classification models used in BCI [9,10]. This variability has been strongly linked to the phenomenon known as BCI illiteracy, where a substantial subset of users is unable to gain intentional control of the system, even after repeated training sessions [11,12]. Beyond its technical implications, this limitation poses a fundamental challenge to the inclusion and scalability of BCI technologies in real-world clinical settings, where system adaptability to diverse neurophysiological profiles is essential [13]. Another key obstacle in the development of EEG-based BCI systems is their limited spatial resolution, which is inherently constrained by volume conduction effects. Unlike imaging modalities such as functional magnetic resonance imaging (fMRI) or magnetoencephalography (MEG), EEG suffers from spatial distortions because neural electrical signals must traverse multiple layers with varying conductivities—such as the skull and scalp—before being recorded at the surface electrodes [14]. This biophysical phenomenon results in signal mixing across electrodes, making it difficult to accurately localize cortical activity [14]. Consequently, spatial specificity is reduced in applications that require fine-grained identification of motor or sensory regions, ultimately limiting the system’s performance in rehabilitation, neurofeedback, and precision control tasks [15].

In this context, various classical signal processing and machine learning methods have been proposed to improve signal quality and extract discriminative features from EEG recordings, particularly in MI paradigms. Among the earliest approaches, time–frequency domain techniques such as the short-time Fourier transform (STFT) [16] and the wavelet transform [17,18] have proven effective in decomposing EEG signals into more informative components by capturing their non-stationary nature. These tools facilitate the identification of relevant brain rhythms, particularly the μ (8–12 Hz) and β (13–30 Hz) bands, which have been widely linked to movement execution and motor imagery [19]. Additionally, the filter bank common spatial patterns (FBCSP) approach extends this principle by dividing the EEG into multiple frequency sub-bands and applying the CSP algorithm to each of them, thereby enhancing class discrimination [20]. While these strategies improve the signal-to-noise ratio (SNR), their effectiveness depends on fixed or heuristically defined frequency ranges, which limits their adaptability to inter-subject spectral variability [21]. Furthermore, CSP variants—such as regularized CSP, discriminative CSP, and sparse CSP—attempt to mitigate overfitting and improve generalization, but remain noise-sensitive and often require subject-specific calibration [22]. Crucially, these approaches provide limited robustness to the spatial distortions inherent in EEG, caused by volume conduction, which restricts their ability to resolve cortical sources with precision and thus limits their performance in tasks requiring spatial specificity [23].

Conversely, recent approaches have leveraged deep learning, particularly convolutional neural networks (CNNs), to extract hierarchical representations from raw EEG signals. Architectures such as EEG network (EEGNet) [24], shallow convolutional network (ShallowNet) and deep convolutional network (DeepConvNet) [25], and temporal–channel fusion network (TCFusionNet) [26] have shown potential in MI classification by learning spatial and temporal patterns without the need for manual feature engineering. These models exhibit increased robustness to noise and, in some cases, can implicitly compensate for spatial distortions through convolutional kernels. Building on this foundation, kernel-based regularized EEGNet (KREEGNet) [27] introduces explicit spatial encodings and specialized convolutional kernels to enhance cortical sensitivity, offering a more targeted solution to the spatial resolution limitations of EEG. Nevertheless, the performance of these architectures remains highly dependent on large, high-quality datasets, and deeper networks are particularly prone to overfitting. Moreover, achieving robust model generalization remains a significant challenge, particularly due to high inter-subject variability in EEG patterns. As a result, most approaches still rely on subject-specific calibration or employ transfer learning strategies to adapt models across individuals [28]. Beyond CNNs, cutting-edge research has begun to explore attention mechanisms, transformer architectures, and generative modeling [29]. Transformer-based models have shown strong performance in capturing long-range temporal dependencies and improving generalization across subjects. For instance, convolutional Transformer network (CTNet) [30] leverages multi-head self-attention to dynamically extract discriminative spatial-temporal features. Similarly, spatial–temporal transformer models have demonstrated robustness in multi-scale temporal feature extraction [31]. Complementary generative approaches, including autoencoders and variational autoencoders (VAEs), have been employed for nonlinear denoising and unsupervised feature learning [32,33]; however, these models inadvertently discard class-discriminative information unless properly regularized [33]. More recently, multimodal and diffusion-based transformer models have been proposed to integrate spatial, temporal, and topological EEG dynamics [34], though their architectural complexity and computational demands may limit their application in real-time or clinical settings [32].

In addition to approaches based on local or spatial features, EEG representation through connectivity models has been extensively explored, including spectral, structural, directed, and functional connectivity. Spectral connectivity, based on measures such as coherence and spectral entropy, allows for the capture of phase and power relationships between cortical regions, but may be sensitive to noise and the choice of frequency bands [35]. Structural connectivity, typically derived from neuroimaging techniques such as MRI, is difficult to obtain from EEG and is rarely integrated directly into non-invasive applications [36]. Directed connectivity aims to identify causal relationships between regions using metrics like partial directed coherence or dynamic causal modeling [37]; although it enhances interpretability, it often entails significant computational complexity. Functional connectivity, by contrast, has been the most widely applied in EEG contexts due to its ability to model statistical dependencies. As a recent advancement, Gaussian functional connectivity has been proposed as a more robust representation for capturing nonlinear spectral-domain relationships. This formulation was employed in the kernel cross-spectral functional connectivity network (KCS-FCNet) model [38], which integrates kernelized functional connectivity to enhance class discrimination in motor imagery paradigms. Overall, although various forms of connectivity have been explored in conjunction with deep learning, their direct application has yet to mature to a point that effectively addresses critical challenges such as inter-subject variability and limited spatial resolution [36].

Here, we introduce EEG-GCIRNet—a Gaussian connectivity-driven EEG imaging representation Network. This framework is designed to transform functional connectivity patterns into a robust image-based representation for MI classification using a variational autoencoder. Unlike conventional approaches, our method creates a rich representation by generating topographic maps from Gaussian functional connectivity, which model nonlinear spatial–functional dependencies across brain regions. Our EEG-GCIRNet framework comprises three key stages:–**Image-based encoding:** Gaussian connectivity-based image representations are encoded into a shared latent space that captures complementary spatio-temporal and frequency information, enabling more discriminative and interpretable feature representations.–**Adaptive Multi-objective training:** The model is optimized through a composite loss that jointly enforces reconstruction fidelity, classification accuracy, and latent space regularization. Crucially, this enables an adaptive learning strategy where the model automatically prioritizes representation learning (reconstruction) over classification when faced with noisy signals, thereby enhancing robustness and mitigating inter-subject variability.–**Physiological Interpretability:** The framework’s variational design allows for deep validation of the learned features. By integrating latent space visualization with layer-wise relevance analysis (Grad-CAM++) and functional connectivity patterns, we move beyond black-box predictions to confirm that the model decisions are driven by genuine neurophysiological mechanisms, such as fronto-parietal network integration.

Experimental evaluations on benchmark MI datasets demonstrate that EEG-GCIRNet consistently outperforms state-of-the-art baselines, achieving superior classification accuracy and effectively mitigating “BCI illiteracy” by eliminating low-performance groups. Crucially, this robustness extends to complex multi-class scenarios, where the framework demonstrates statistically significant superiority over advanced temporal-channel fusion architectures. Moreover, interpretability analyses reveal distinct functional connectivity structures that align with known motor cortical regions, offering neurophysiological validation of the model’s learned representations. As such, the proposed framework advances the development of robust and interpretable EEG-based BCIs, paving the way for adaptive neurotechnologies in rehabilitation, assistive communication, and motor recovery. By combining multimodal encoding, multi-objective learning, and explainable AI, EEG-GCIRNet contributes a reproducible and scalable paradigm for addressing two of the most persistent challenges in EEG-based BCI research: inter-subject variability and limited spatial specificity.

The agenda is as follows. Section 2 describes the materials and methods. Section 4 presents the experiments and results. Section 5 provides the discussion. Finally, Section 6 outlines the conclusions, limitations and future work.

## 2. Materials and Methods

### 2.1. GIGAScience Dataset for EEG-Based Motor Imagery

The Giga Motor Imagery–DBIII (GigaScience) dataset, publicly available at http://gigadb.org/dataset/100295 (accessed on 1 July 2025), provides one of the most comprehensive EEG corpora for MI analysis. The dataset comprises recordings from 52 healthy participants (50 with usable data), each performing a single EEG-MI session. Every session consists of five to six experimental blocks, with each block containing approximately 100–120 trials per class. Each trial spans seven seconds and follows a fixed timeline: an initial blank screen (0–2 s), a visual cue indicating either left- or right-hand MI (2–5 s), and a concluding blank interval (5–7 s). Inter-trial intervals vary randomly between 0.1 s and 0.8 s to mitigate anticipatory bias, as illustrated in Figure 1. EEG signals were recorded at a sampling frequency of 512 Hz using a 64-channel cap arranged according to the international 10–10 electrode placement system. In addition to the MI sessions, the dataset includes recordings of real motor execution and six auxiliary non-task-related events—eye blinks, vertical and horizontal eye movements, head motions, jaw clenching, and resting state—enabling a broader exploration of EEG noise sources and artifact correction. This multimodal composition makes the GigaScience particularly valuable for benchmarking advanced deep learning and connectivity-based EEG decoding frameworks, as it supports both intra-subject and inter-subject generalization studies.

### 2.2. EEG Motor Movement/Imagery Database

The EEG Motor Movement/Imagery Database (EEGMMIDB) [39], available at https://physionet.org/content/eegmmidb/1.0.0/ (accessed on 7 December 2025), was employed as a secondary benchmark to evaluate the framework’s versatility. This dataset comprises electroencephalographic recordings from 109 healthy participants performing a variety of real and imagined motor tasks. The EEG signals were acquired using 64 scalp electrodes positioned according to the international 10–10 system and sampled at 160 Hz. The recording protocol encompasses 14 sessions per subject, including resting-state conditions, motor execution tasks, and MI tasks. Data segmentation was performed based on event annotations to define discrete trials. Each trial corresponds to a 4.1 s window, yielding a raw input matrix of 64channels×640samples. To standardize the classification task, the original PhysioNet labels (0–11) were reorganized into five interpretable MI categories: (1) Right Hand (labels 0–1); (2) Left Hand (labels 2–3); (3) Both Hands (labels 4–5); (4) Both Feet (labels 6–7); and (5) Rest (labels 8–9). Annotations 10 and 11, corresponding to non-motor tasks, were discarded. In alignment with the experimental design of the GigaScience dataset, we focused exclusively on the MI-related sessions, excluding motor execution trials. The experimental configuration is summarized in Figure 2.

Each MI trial is structured for the proposed framework. Let $\left\{X\in R^{C\times \tau },\;y\in {\left\{0,1\right\}}^{Q}\right\}$ denote a multichannel EEG and its associated MI target label, where X represents the EEG recording with *C* spatial channels and τ temporal samples, and y is a one-hot encoded vector indicating the MI class among *Q* possible categories.

### 2.3. Laplacian Filtering and Time Segmentation

To enhance the spatial resolution and mitigate the volume conduction effects inherent in EEG recordings, a Surface Laplacian filter is applied to each trial X. This filter acts as a spatial high-pass filter by estimating the second spatial derivative of the scalp potential at each electrode c∈C with respect to its neighbors $c^{\prime }\in C$, where $c\neq c^{\prime }$. Following the methodology in [40], this is achieved by using spherical splines to project the electrode positions onto a unit sphere, which allows for the interpolation of scalp potentials via Legendre polynomials. The interaction between any pair of electrodes $\left(c,c^{\prime }\right)$ is modeled as:(1)$$p\left(c,c^{\prime }\right)=\frac{1}{4\pi }\sum_{n=1}^{N_{max}}\frac{\alpha \left(2n+1\right)P_{n}\left(\mathrm{cosdist}(e_{c},e_{c^{\prime }})\right)}{{\left(n\left(n+1\right)\right)}^{\rho -\alpha }},$$
where $P_{n}$ is the Legendre Polynomial of order *n*, $N_{max}$ is the highest polynomial order considered, $\rho \in R^{+}$ is a smoothness constant, and $e_{c},e_{c^{\prime }}\in R^{3}$ are the 3D electrode positions normalized to a unit-radius sphere. The cosine distance is defined as $\mathrm{cosdist}\left(e_{c},e_{c^{\prime }}\right)=1-{∥e_{c}-e_{c^{\prime }}∥}_{2}/2$.

The Laplacian-filtered EEG data, denoted as $X_{L}\in R^{C\times \tau }$, is subsequently computed using the weighting matrices derived from the spline interpolation:(2)$$X_{L}=H{\left(X^{⊤}G_{s}^{-1}-X^{⊤}G_{s}^{-1}1G_{s}^{-1}/1^{⊤}G_{s}^{-1}1\right)}^{⊤},$$
where $1\in R^{C}$ is a column vector of ones, $I\in R^{C\times C}$ is the identity matrix, and λ∈[0,1] is a regularization parameter. The matrix $G_{s}=G+\lambda I$ is a regularized (smoothed) version of G. The weighting matrices $G,H\in R^{C\times C}$ hold the elements derived from Equation (Equation 1), with their specific values determined by the parameter α as follows:(3)$$\alpha =\left\{\begin{array}{cc} 1, & \mathrm{where}\;p\left(c,c^{\prime }\right)=g\left(c,c^{\prime }\right)\\ -1, & \mathrm{where}\;p\left(c,c^{\prime }\right)=h\left(c,c^{\prime }\right) \end{array}\right.,$$
where $g(c,c^{\prime })$ and $h(c,c^{\prime })$ are the elements of matrices G and H, respectively. This filtering step produces a spatially enhanced representation $X_{L}$ that serves as input for the subsequent feature extraction stages.

Further, to focus the analysis exclusively on the period of active motor imagery, the Laplacian-filtered signal $X_{L}$ is temporally segmented. The time window corresponding to the MI task, specifically between $t_{s}$ and $t_{e}$ seconds of each trial, is retained. Let $t_{s}$ and $t_{e}$ be the start and end times of the MI segment, and $f_{s}$ be the sampling frequency, the segmented signal $X_{seg}\in R^{C\times \tau }$ is obtained as:(4)$$X_{seg}=X_{L}\left[:,t_{s}\cdot f_{s}:t_{e}\cdot f_{s}\right],$$
where the slicing notation $\left[t_{s}\cdot f_{s}:t_{e}\cdot f_{s}\right]$ indicates the selection of temporal samples from the start index to the end index. For brevity, this segmented signal will be denoted as X in the subsequent sections.

### 2.4. Kernel-Based Cross-Spectral Gaussian Connectivity for EEG Imaging

To model the mutual dependency between EEG channels, we consider any two channels $x_{c},x_{c^{\prime }}\in R^{\tau }$ from a given trial X (where $c,c^{\prime }\in C$). Their mutual dependency can be captured using a stationary kernel $\kappa :R^{\tau }\times R^{\tau }\to R$, which maps both signals into a reproducing kernel Hilbert space (RKHS) via a nonlinear feature map $\phi :R^{\tau }\to H$ [41]. Indeed, according to Bochner’s theorem, a sufficient condition for the kernel κ to be stationary is that it admits a spectral representation [42]:(5)$$\kappa \left(x_{c}-x_{c^{\prime }}\right)=\int_{\Omega_{b}}\exp\left(j2\pi {\left(x_{c}-x_{c^{\prime }}\right)}^{⊤}f\right)S_{x_{c}x_{c^{\prime }}}\left(f\right)\;df,$$
where $f\in \Omega \subseteq R^{\tau }$ is a frequency vector, and $S_{x_{c}x_{c^{\prime }}}\left(f\right)=\frac{dP_{x_{c}x_{c^{\prime }}}\left(f\right)}{df}\in C$ is the cross-spectral density between x and $x^{\prime }$, derived from the spectral distribution $P_{x_{c}x_{c^{\prime }}}\left(f\right)$.

Building on this spectral representation, the cross-spectral power within a specific frequency band $\Omega_{b}$ can be computed via the Fourier transform of the kernel:(6)$$P_{x_{c}x_{c^{\prime }}}\left(\Omega_{b}\right)=2\int_{\Omega_{b}}F\left\{\kappa (x_{c}-x_{c^{\prime }})\right\}df,$$
where F{·} denotes the Fourier transform. This spectral formulation allows capturing both linear and nonlinear dependencies in the frequency domain, making it particularly useful for analyzing brain signals.

A widely used choice for κ is the Gaussian kernel, which ensures smoothness, locality, and analytic tractability [43]:(7)$$\kappa_{G}\left(x_{c}-x_{c^{\prime }};\sigma \right)=\exp\left(-\frac{∥x_{c}-x_{c^{\prime }}{∥}_{2}^{2}}{2\sigma^{2}}\right),$$
where $\sigma \in R^{+}$ is a bandwidth hyper-parameter.

Inspired by the kernel-based spectral approaches introduced in [38,44], we compute a Gaussian kernel cross-spectral connectivity estimator to encode spatio–frequency interactions among pairwise EEG channels. Specifically, for each EEG channel *c*, a band-limited spectral reconstruction is obtained as:(8)$$x_{c}^{\left(\Omega_{b}\right)}=F^{-1}\left\{F\left\{x_{c}\right\};\Omega_{b}\right\},$$
where $\Omega_{b}$ denotes a given frequency bandwidth (rhythm). Then, the Gaussian Function Connectivity (GFC) matrix $K^{\left(\Omega_{b}\right)}\in {\left[0,1\right]}^{C\times C}$ is derived to quantify the degree of similarity between the spectral representations of all channel pairs, as:(9)$$K_{cc^{\prime }}^{\left(\Omega_{b}\right)}=\kappa_{G}\left(x_{c}^{\left(\Omega_{b}\right)}-x_{c^{\prime }}^{\left(\Omega_{b}\right)};\sigma_{\Omega_{b}}\right),$$
where $\kappa_{G}\left(\cdot ;\sigma_{\Omega_{b}}\right)$ is a Gaussian kernel with scale parameter $\sigma_{\Omega_{b}}$. To ensure adaptive sensitivity across rhythms, $\sigma_{\Omega_{b}}$ is estimated as the median of all pairwise Euclidean distances between spectral reconstructions $x_{c}^{\left(\Omega_{b}\right)}$ and $x_{c^{\prime }}^{\left(\Omega_{b}\right)}$, $\forall \;c,c^{\prime }\in C$. This formulation provides a data-driven normalization of connectivity strength, enabling robust comparison across heterogeneous EEG rhythms and subjects.

Afterward, we propose to compute an EEG connectivity flow from $K^{\left(\Omega_{b}\right)}$, preserving a direct one-to-one correspondence with the electrode spatial configuration. Specifically, the GFC flow vector $g^{\left(\Omega_{b}\right)}\in {\left[0,1\right]}^{C}$ holds elements:(10)$$g_{c}^{\left(\Omega_{b}\right)}=\frac{1}{C}\sum_{\begin{array}{c} c^{\prime }=1\\ c^{\prime }\neq c \end{array}}^{C}K_{cc^{\prime }}^{\left(\Omega_{b}\right)},$$
where each element $g_{c}^{\left(\Omega_{b}\right)}$ represents the mean functional coupling of channel *c* with all other channels within the given frequency band $\Omega_{b}$. The latter compresses the pairwise connectivity information into a compact, channel-wise flow representation while retaining the spatial and spectral data patterns.

To ensure a consistent feature scale for the imaging stage, the GFC flow vectors are normalized across the entire training dataset. Specifically, a channel-wise Min-Max normalization is applied, scaling the connectivity values of each channel to a uniform range of [0,1]. This procedure preserves the relative topography of neural connectivity while standardizing the input scale, yielding the normalized flow vector ${\tilde{g}}^{\left(\Omega_{b}\right)}$ for each trial.

### 2.5. Topographic Map Generation

The final feature engineering step transforms the one-dimensional GFC flow vectors into two-dimensional topographic images, creating a data representation suitable for convolutional neural network (CNN) architectures. For each trial, this process converts the normalized flow vector ${\tilde{g}}^{\left(\Omega_{b}\right)}\in {\left[0,1\right]}^{C}$ from each frequency band into a corresponding topographic image $T^{\left(\Omega_{b}\right)}\in R^{\tilde{H}\times \tilde{W}}$.

This transformation is accomplished via spatial interpolation guided by Delaunay triangulation. First, the set of 2D scalp coordinates of the electrodes, $P={\left\{\left(x_{c},y_{c}\right)\right\}}_{c=1}^{C}$, is triangulated. This partitions the electrode layout into a mesh of non-overlapping triangles, where the circumcircle of each triangle contains no other electrode points. This triangulation provides a structured grid for interpolating the connectivity values, where each element ${\tilde{g}}_{c}^{\left(\Omega_{b}\right)}$ is associated with its corresponding coordinate $\left(x_{c},y_{c}\right)$.

To generate the final image, the value V(x,y) for each pixel is computed using barycentric interpolation within its enclosing triangle $T_{k}=\left(p_{1},p_{2},p_{3}\right)$:(11)$$V\left(x,y\right)=\lambda_{1}\left(x,y\right)v\left(p_{1}\right)+\lambda_{2}\left(x,y\right)v\left(p_{2}\right)+\lambda_{3}\left(x,y\right)v\left(p_{3}\right),$$
where $v(p_{i})$ is the connectivity value at vertex $p_{i}$, and $\lambda_{1},\lambda_{2},\lambda_{3}$ are the barycentric coordinates of (x,y) satisfying $\sum_{j=1}^{3}\lambda_{j}=1$ and $\lambda_{j}\geq 0$. This procedure is applied across all pixels to render the smooth topographic map $T^{\left(\Omega_{b}\right)}$. The resulting set of four maps (one for each frequency band) is then stacked to form a multi-channel image, which serves as the final input to the deep learning model.

### 2.6. EEG-GCIRNet: Multimodal Architecture

The proposed model is a variational autoencoder (VAE) designed to process topographic maps derived from functional connectivity representations. Its architecture relies on a single input stream that learns to extract and encode the most relevant spatial features from the maps, which are then projected into a shared latent space where a multivariate Gaussian distribution is modeled. From this latent space, the model simultaneously performs reconstruction of the topographic maps and classification of motor imagery tasks. These objectives are integrated within a composite loss function that balances reconstruction fidelity, classification accuracy, and latent space regularization. This approach enables the learning of robust and interpretable latent representations capable of capturing discriminative spatial relationships and adapting to the variability and noise inherent in EEG signals.

The core of our framework is a VAE based on the LeNet-5 architecture, which is partitioned into three functional blocks: an encoder, a decoder, and a classifier, all operating on the shared latent space. Let $Y\in R^{\tilde{H}\times \tilde{W}\times B}$ be the multi-channel input image for a given trial, formed by stacking the *B* topographic maps (one for each frequency band).

The encoder, defined as a function $E_{\phi }$ parameterized by ϕ, maps the input image *Y* to the parameters of the posterior distribution $q_{\phi }\left(z|Y\right)$. This transformation is realized through a composition of functions, where each function represents a layer in the network:(12)$$h_{E}=\left(f_{5}\circ f_{4}\circ f_{3}\circ f_{2}\circ f_{1}\right)\left(Y\right)$$

Here, $f_{1}$ and $f_{3}$ are convolutional layers with SELU activation, $f_{2}$ and $f_{4}$ are average pooling layers, and $f_{5}$ is a fully connected layer with SELU activation after flattening the feature maps. The resulting hidden representation $h_{E}$ is then linearly transformed to produce the mean vector μ and the log-variance vector $\log\sigma^{2}$ of the latent space:(13)$$\begin{array}{cc} \;\;\mu & =W_{\mu }h_{E}+b_{\mu } \end{array}$$(14)$$\begin{array}{cc} \log\sigma^{2} & =W_{\sigma }h_{E}+b_{\sigma } \end{array}$$

The decoder, defined as a function $D_{\theta }$ parameterized by θ, reconstructs the original input image $\hat{Y}$ from a latent vector $z∼q_{\phi }\left(z|Y\right)$. Its architecture mirrors the encoder by composing functions that progressively up-sample the representation to the original image dimensions:(15)$$\hat{Y}=\left(g_{3}\circ g_{2}\circ g_{1}\right)\left(z\right)$$

Here, $g_{1}$ is a fully connected layer followed by a reshape operation, $g_{2}$ is a transposed convolutional layer with SELU activation, and $g_{3}$ is a final transposed convolutional layer with a Sigmoid activation to ensure the output pixel values are in a normalized range.

Concurrently, the classifier, a function $C_{\psi }$ parameterized by ψ, predicts the MI task label probabilities $\hat{p}$ from the same latent vector *z*. It is implemented as a multi-layer perceptron:(16)$$\hat{p}=C_{\psi }\left(z\right)=\mathrm{Softmax}\left(W_{c}\left(\mathrm{SELU}\left(W_{h}z+b_{h}\right)\right)+b_{c}\right)$$

Formally, the latent vector $z_{i}$ for a given input sample *i* is computed using the reparameterization trick:(17)$$z_{i}=\mu_{i}+\exp\left(\frac{1}{2}\log\sigma_{i}^{2}\right)⊙\epsilon_{i},\;\epsilon_{i}∼N\left(0,I\right)$$
where $\mu_{i}$ and $\sigma_{i}^{2}$ denote the mean and variance of the approximate posterior distribution learned by the encoder for sample *i*, and $\epsilon_{i}$ is drawn from a standard normal distribution. The exponential term ensures the sampled standard deviation remains strictly positive.

The total objective function, $L_{\mathrm{total}}$, is defined as a weighted sum of the three loss terms:(18)$$\begin{array}{cc} L_{\mathrm{total}}=\; & \lambda_{\mathrm{REC}}\cdot L_{\mathrm{REC}}+\lambda_{\mathrm{CLA}}\cdot L_{\mathrm{CLA}}+\lambda_{\mathrm{REG}}\cdot L_{\mathrm{REG}} \end{array}$$
where(19)$$\begin{array}{cc} L_{\mathrm{REC}} & =\frac{\frac{1}{N}\sum_{i=1}^{N}{∥Y_{i}-{\hat{Y}}_{i}∥}_{F}^{2}}{\frac{1}{N}\sum_{i=1}^{N}{∥Y_{i}-\bar{Y}∥}_{F}^{2}} \end{array}$$(20)$$\begin{array}{cc} \;\;\;\;L_{\mathrm{CLA}} & =\frac{1}{N}\sum_{i=1}^{N}\frac{L_{\mathrm{cce}}\left(p_{i},{\hat{p}}_{i}\right)}{L_{\mathrm{cce}}\left(\left[0.5,0.5\right],{\hat{p}}_{i}\right)} \end{array}$$(21)$$\begin{array}{cc} \;\;\;\;\;\;\;\;\;\;L_{\mathrm{REG}} & =\frac{1}{N\log N}\sum_{i=1}^{N}D_{\mathrm{KL}}(q_{\phi }\left(z_{i}|Y_{i}\right)||N\left(0,I\right)) \end{array}$$
where $\lambda_{\mathrm{REC}}$, $\lambda_{\mathrm{CLA}}$, and $\lambda_{\mathrm{REG}}$ are hyperparameters controlling the contribution of each term. The first component, $L_{\mathrm{REC}}$, is the normalized mean squared error (NMSE), which evaluates reconstruction accuracy by comparing the original topographic maps ($Y_{i}$) and their reconstructions (${\hat{Y}}_{i}$), normalized by the dataset’s variance ($\bar{Y}$ is the mean image). The Frobenius norm, ${∥\cdot ∥}_{F}$, is used for the image-wise error. The second term, $L_{\mathrm{CLA}}$, represents the normalized binary cross-entropy (NBCE). It penalizes misclassifications between the true one-hot labels ($p_{i}$) and predicted probabilities (${\hat{p}}_{i}$), while adjusting the loss based on the entropy over an ideal, non-informative prediction (e.g., a uniform distribution [0.5,0.5]). This maintains a balanced contribution from all classes. Finally, the third term, $L_{\mathrm{REG}}$, is the normalized Kullback–Leibler (KL) divergence between the approximate posterior $q_{\phi }\left(z_{i}|Y_{i}\right)$ and a unit Gaussian prior N(0,I). This regularizes the latent space, promoting smoothness and disentanglement in the learned representations.

This KL divergence term encourages the latent representations to follow a standard normal distribution, promoting structure and generalization. The use of logN in the denominator prevents this term from dominating the loss in large batches. Taken together, these components ensure that the model jointly optimizes for faithful reconstruction, discriminative performance, and a well-regularized latent structure—crucial for interpretable and generalizable multimodal brain–computer interfaces.

The model is trained by solving the following optimization problem:$$\left\{\phi^{*},\theta^{*},\psi^{*}\right\}=\arg\min_{\phi ,\theta ,\psi }L_{\mathrm{total}}$$
where {ϕ,θ,ψ} denotes the complete set of trainable parameters in the encoder, decoder, and classifier, respectively.

## 3. Experimental Set-Up

This work presents EEG-GCIRNet, a framework for MI classification built upon topographic maps derived from functional connectivity. The proposed methodology, illustrated in Figure 3, comprises three primary stages: (i) preprocessing raw EEG signals to compute GFC-based flow vectors; (ii) generating 2D topographic maps from these vectors; and (iii) processing the resulting images with a deep learning architecture for simultaneous classification and reconstruction.

### 3.1. Stage 1: Signal Preprocessing and Feature Engineering

First, an average reference was applied, which included the original reference electrode to ensure the data retained full rank. Subsequently, a fifth-order Butterworth bandpass filter was applied in the 4–40 Hz range. To reduce computational load and maintain consistency across the evaluated deep learning models, the filtered signals of GigaScience and EEGMMIDB dataset were resampled from 512 Hz and 160 Hz to 128 Hz, respectively [27,38]. This study evaluates the proposed framework on two distinct experimental scenarios. First, for the GigaScience dataset, we focus on the binary classification of Left Hand versus Right Hand motor imagery (Q=2). This analysis was conducted on a subset of 50 subjects from the original cohort, with participants 29 and 34 excluded due to data availability constraints. Second, for the EEGMMIDB database, the task is extended to the multi-class classification of five interpretable MI categories (Q=5). To ensure rigorous experimental consistency and following the experimental setup suggested in [45], a specific subset of ten subjects {3, 7, 8, 9, 12, 13, 40, 41, 49, 50} was selected, incorporating all valid trials associated with the target classes for these participants.

Building upon this preprocessed data, the feature engineering process involves applying a Surface Laplacian filter to enhance the spatial resolution of the signals. Specifically, the temporal segmentation was adapted to the recording protocol of each dataset. For the GigaScience repository, the data was segmented to isolate the active MI period, retaining the window from 2.5 to 4.5 s. In contrast, for the EEGMMIDB collection, the entire trial duration of 4.1 s was utilized. The segmented data is further decomposed via band-pass filtering into four functionally distinct frequency bands: μ (8–12 Hz), low-beta ($\beta_{l}$, 12–15 Hz), mid-beta ($\beta_{m}$, 15–20 Hz), and high-beta ($\beta_{h}$, 18–40 Hz). For each frequency band, GFC is computed to quantify the functional relationships between all channel pairs. The length scale hyperparameter $\sigma \in R^{+}$, ruling the variance of the described data is adjusted to their median estimate as performed in [46]. The resulting connectivity information is then condensed into a normalized, channel-wise flow vector for each band.

### 3.2. Stage 2: Topographic Map Generation

The second stage of the pipeline transforms the one-dimensional, GFC-based flow vectors into a two-dimensional, image-based representation suitable for processing with a CNN. This conversion is critical as it re-introduces the spatial topography of the EEG electrodes, allowing the model to learn spatially coherent features.

This transformation was achieved via spatial interpolation, a process implemented using the visualization utilities within the MNE-Python library (https://mne.tools/stable/index.html, accessed on 1 June 2025). For each frequency band, the corresponding flow vector’s values are mapped to the 2D coordinates of the EEG electrodes. A mesh is then constructed over these coordinates using Delaunay triangulation. The pixel values for the final topographic map are subsequently estimated using linear barycentric interpolation within this mesh. This procedure is repeated for each of the four frequency bands (μ, $\beta_{l}$, $\beta_{m}$, and $\beta_{h}$), yielding a set of four distinct topographic maps per trial. These maps are then stacked along the channel dimension to form a single, multi-channel image of size 40×40×4. This resulting data structure serves as the final input to the EEG-GCIRNet architecture, providing a rich, spatio-spectral representation of the brain’s functional connectivity during motor imagery.

### 3.3. Stage 3: EEG-GCIRNet Architecture and Training

The core of this model is a VAE with a convolutional architecture inspired by LeNet-5. This architecture is composed of three interconnected functional blocks operating on a shared latent space: an encoder, a decoder, and a classifier.

The encoder block consists of two sequential pairs of convolutional and average pooling layers, which extract hierarchical spatial features from the input image. These features are then flattened and passed through a dense layer to produce a compact representation, which in turn parameterizes the mean (μ) and log-variance ($\log\sigma^{2}$) vectors of the latent space. The decoder mirrors this structure using transposed convolutional layers to upsample the latent representation back to the original image dimensions. Concurrently, the classifier, a simple multi-layer perceptron, operates on the same latent vector to perform the final classification. The detailed layer-wise configuration of the EEG-GCIRNet is summarized in Table 1.

The EEG-GCIRNet model was trained end-to-end by optimizing the composite loss function described in Section 2.6. The training was performed using the Adam optimizer with an initial learning rate of $1\times 10^{-3}$. The hyperparameters that weight the loss components (i.e., $\lambda_{\mathrm{REC}}$, $\lambda_{\mathrm{CLA}}$, and $\lambda_{\mathrm{REG}}$) were set using KerasTuner framework to ensure a balanced contribution from reconstruction, classification, and regularization objectives during training, subject to the constraint that they sum to one. The model was trained for a total of 200 epochs with a batch size of 64. An early stopping mechanism was employed with a patience of 10 epochs, monitoring the validation loss to prevent overfitting and save the model with the best generalization performance. The model performance was evaluated using a subject-specific validation strategy. For each of the considered subjects, we employed a *Stratified Shuffle Split* cross-validation scheme with 5 repetitions (n_splits = 5). In each iteration, the trials were randomly partitioned into 80% for training and 20% for testing (test_size = 0.2), ensuring that the class distribution remained balanced in both sets. The results reported in this study correspond to the average accuracy obtained strictly on the test sets across these 5 splits, ensuring that the reported metrics reflect the model’s generalization capability on unseen data.

### 3.4. Evaluation Criteria

The performance of the proposed EEG-GCIRNet was rigorously evaluated using a multi-faceted approach tailored to the specific requirements of each experimental scenario. For the GigaScience dataset (Q=2), the primary quantitative metric was subject-specific binary classification accuracy, benchmarked against seven baseline and state-of-the-art models. For the EEGMMIDB dataset, the evaluation was extended to a multi-class setting (Q=5), focusing on the model’s ability to resolve complex decision boundaries across five distinct motor imagery categories.

To validate the findings, a robust statistical framework was employed. A Friedman test was used to assess overall significance across models, followed by post-hoc pairwise *t*-tests for direct comparisons. To account for the multiple comparisons problem, the resulting *p*-values were adjusted using the *Holm-Bonferroni* correction method to control the family-wise error rate. Furthermore, the framework’s robustness and generalization capabilities were analyzed by stratifying the GigaScience cohort into “Good” (accuracy >80%), “Mid” (accuracy 60–80%), and “Bad” (accuracy <60%) performance groups based on the standard EEGNet baseline. This stratification allowed for a targeted assessment of the model’s corrective effectiveness across varying EEG signal qualities.

Beyond quantitative metrics, the evaluation delved into the model’s interpretability by leveraging its variational autoencoder architecture and gradient-based analysis. This qualitative assessment involved three key methods: (i) a visual analysis of the reconstructed topographic maps to confirm that the model learned physiologically relevant spatio-spectral patterns; (ii) the visualization of the latent space using t-SNE projections to directly inspect the quality of class separability and feature disentanglement; and (iii) a layer-wise relevance analysis using Grad-CAM++. The latter was employed to verify that the model’s decision-making is driven by spatial contributions from neurophysiologically valid regions—such as the sensorimotor cortex—rather than artifacts. This combined quantitative and qualitative evaluation provides a holistic validation of the EEG-GCIRNet framework, covering its accuracy, statistical significance, and the meaningfulness of its learned internal representations.

## 4. Results

### 4.1. Binary MI Classification Performance on the GigaScience Database

To evaluate the model’s robustness against the well-documented challenge of “BCI illiteracy”, we stratified subjects into performance groups. This stratification was based on the accuracy of EEGNet, a widely-used, state-of-the-art benchmark for EEG-based BCI. This approach allows for a standardized and unbiased assessment of our model’s comparative effectiveness, particularly its ability to improve performance for users who struggle with conventional systems. Figure 4 illustrates the subject-wise classification performance, where a clear advantage of EEG-GCIRNet becomes evident, particularly in challenging cases. Within the “Bad” group—comprising subjects with low-quality or highly variable EEG signals—conventional models like CSP, ShallowConvNet, and DeepConvNet consistently yield low and unstable results, reflecting their limited ability to handle noise and inter-subject variability. While architectures such as EEGNet and KREEGNet show more stable behavior, their performance remains inconsistent. In stark contrast, EEG-GCIRNet entirely eliminates the “Bad” performance group, demonstrating a generalized improvement and a notable reduction in inter-subject variability. This outcome strongly suggests that the model’s variational formulation and latent space regularization provide robust feature encoding, effectively preventing the critical performance failures seen in other architectures [47,48].

EEG-GCIRNet extends its advantage into the “Mid” group, consistently outperforming competing models like TCFusionNet and KREEGNet across most subjects. This stability under intermediate conditions underscores the model’s strong generalization capability, as it delivers reliable performance even with moderately variable EEG signals. These results validate the effectiveness of the variational approach in preserving discriminative information while maintaining training stability [49].

In the “Good” group, where most architectures achieve high accuracy, EEG-GCIRNet performs competitively, matching or exceeding the results of TCFusionNet, KREEGNet, and EEGNet. Critically, its performance is marked by greater consistency across subjects, highlighting its ability to maintain high accuracy without overfitting. This behavior contrasts sharply with deeper architectures like DeepConvNet, which are more susceptible to performance degradation in subject-specific tasks [15].

Collectively, the subject-wise results in Figure 4 underscore that EEG-GCIRNet achieves a superior balance of accuracy, stability, and generalization. This positions it as a highly effective and well-rounded model for EEG decoding [50,51]. The aggregate performance metrics summarized in Table 2 confirm these subject-wise trends. EEG-GCIRNet stands out as the best-performing model overall, achieving the highest average accuracy (81.82%) and the lowest standard deviation (±10.15), which confirms its strong generalization and inter-subject stability.

Notably, while KREEGNet achieves a competitive average accuracy (77.32%), its greater performance variability (±14.74) indicates reduced stability. Taken together, these results position EEG-GCIRNet as the most reliable alternative, outperforming all reference architectures in both classification accuracy and consistency.

To validate that the observed performance differences among the models were statistically meaningful, a rigorous statistical analysis was conducted. A Friedman test was first applied to the subject-wise accuracies of all eight models, yielding a test statistic with a *p*-value lower than the detection threshold (p<0.01) for proposed EEG-GCIRNet model. This result allows for the rejection of the null hypothesis of equal medians with a high level of confidence, confirming that statistically notable differences exist across the evaluated architectures.

The nature of these differences is detailed in Figure 5 and Table 3. The subject-wise ranking heatmap (Figure 5a) visually confirms the performance tiers, highlighting the consistent top rankings of EEG-GCIRNet and KREEGNet, the intermediate performance of models like TCFusionNet and ShallowConvNet, and the instability of DeepConvNet and CSP. The matrix of *p*-values from post-hoc pairwise *t*-tests (Figure 5b) further reinforces this, revealing statistical differences between the high- and low-performing models.

The summary of these statistical measures in Table 3 provides a conclusive overview. The average *p*-values shown for each model represent the mean of its corrected *p*-values from the pairwise comparisons against all other models. EEG-GCIRNet emerges as the clear top-performing model, securing the lowest average ranking (2.32) and the only average *p*-value indicating notable statistical significance (p=0.007). This result demonstrates a robust and consistent performance advantage over the other architectures. These findings align perfectly with the accuracy results from Table 2, cementing its reliability as a unimodal image-based model for MI classification.

While the KREEGNet model is its closest competitor with an average ranking of 2.48, its average *p*-value (0.07) and greater performance variability indicate a lack of statistical significance and low stability compared to EEG-GCIRNet. This consistent advantage is likely attributable to EEG-GCIRNet’s variational formulation, which promotes more uniform representations across subjects. In summary, the statistical analysis positions EEG-GCIRNet as the most prominent model in terms of both accuracy and statistical significance.

### 4.2. Mitigation of BCI Illiteracy and Cross-Subject: Bi-Class Scenario

A key advantage of EEG-GCIRNet lies in its ability to generalize across subjects and its robustness to the high inter-subject variability inherent in EEG data. To assess this, a stratified analysis was performed based on signal quality, with the performance distributions for each model shown in Figure 6.

In high-quality signal conditions (“Good” group, Figure 6a), all methods perform well (70–100% accuracy). However, EEG-GCIRNet is distinguished by its more compact distribution concentrated at the upper end of the accuracy range, indicating superior inter-subject stability and performance consistency. This behavior, likely stemming from its latent space regularization, contrasts with the wider distributions of models like DeepConvNet and CSP, which reflect greater variability. This advantage becomes more pronounced in the “Mid” group (Figure 6b), which represents subjects with moderate signal quality. While most architectures exhibit broader and less stable performance distributions, EEG-GCIRNet maintains a distribution centered around high accuracy values (70–85%) with only moderate dispersion. This demonstrates its ability to preserve generalization and deliver stable performance even as class separability decreases.

The most compelling evidence of the model’s robustness is found in the “Bad” group (Figure 6c), which reflects the most challenging EEG conditions. Here, most models exhibit broad distributions shifted toward low accuracies (40–70%). Crucially, EEG-GCIRNet is the only model for which this group is empty, reaffirming its resilience against signal degradation and its superior stability, as previously suggested by the global accuracy and ranking analyses.

These distributional advantages translate directly into substantial accuracy gains, as summarized in Table 4. The improvements are most pronounced for subjects who perform poorly with baseline models. In the “Bad” group, where EEGNet achieves an average accuracy of only 54.65%, EEG-GCIRNet provides a remarkable ∼22% increase, elevating the average performance to 76.20%. A substantial gain of ∼14% is also observed in the “Mid” group. Conversely, for the “Good” group, where the baseline performance is already high, EEG-GCIRNet maintains a comparable accuracy with only a slight decrease (∼2%). This confirms that the significant gains for challenging cases are not achieved at the expense of performance in high-quality signal conditions.

Figure 7 illustrates the subject-specific improvements of EEG-GCIRNet over the EEGNet baseline, providing direct visual evidence of our framework’s corrective capability. Since the performance groups were stratified based on EEGNet’s accuracy, the figure demonstrates our model’s ability to significantly elevate the performance of users who struggle with conventional BCI systems, thereby directly addressing the challenge of BCI illiteracy. While most participants benefit from moderate accuracy gains, a notable subset—including subjects 21, 40, 24, 30, 42, 9, 15, and 7 exhibit a pronounced transition from “Bad” to “Good” performance levels. For several of these individuals (e.g., subjects 24, 9, and 7), accuracy exceeds the 80% threshold. This performance leap reinforces the hypothesis that latent space modeling in EEG-GCIRNet not only mitigates the limitations of signal noise but also provides an effective mechanism for uncovering discriminative structure in signals previously deemed uninformative.

These findings are highly relevant for developing inclusive BCI systems. The fact that several initially low-performing individuals achieve high accuracy suggests that EEG-GCIRNet can serve a corrective function within BCI pipelines, enhancing usability and consistency. This benefit extends to “Mid-performing” users as well, many of whom are elevated to the high-performing category. This corrective and enhancing capability supports the use of latent generative models in real-world BCI contexts, where adaptability and generalization are critical [56,57,58].

### 4.3. Validation on Complex Scenarios: Multi-Class Decoding on EEGMMIDB Collection

To assess the scalability of the proposed framework beyond binary classification, we evaluated its performance on the EEGMMIDB dataset, which involves a challenging 5-class motor imagery task. This scenario requires the model to disentangle multiple distinct neural patterns (Left Hand, Right Hand, Both Hands, Both Feet, and Rest), significantly increasing the complexity of the decision boundaries compared to the GigaScience benchmark.

Table 5 presents the comparative results against a curated set of state-of-the-art architectures. These baselines were selected to represent the broad spectrum of current deep learning strategies for EEG decoding: EEGNet serves as the compact, general-purpose benchmark; DeepConvNet and ShallowConvNet represent established end-to-end architectures that optimize temporal and spatial filters directly from raw data; and TCFusionNet is included to represent recent advancements in complex temporal-channel fusion mechanisms. Against this diverse backdrop of raw signal decoders, EEG-GCIRNet achieves the highest average accuracy of 75.20±4.63%, surpassing all baseline models by a considerable margin. While reference architectures such as TCFusionNet and DeepConvNet plateau around ∼68% (with 68.89% and 68.35% respectively), our approach yields a net performance gain of approximately 6.3%. This superiority is particularly notable given the difficulty of the task, suggesting that the connectivity-driven topographic maps provide a richer feature space than raw temporal signals for distinguishing between spatially overlapping classes, such as “Both Hands” versus single-hand imagery.

The robustness of these findings is visually confirmed in Figure 8, which details subject-wise accuracy sorted by baseline performance. EEG-GCIRNet maintains a strict performance advantage across the entire cohort, with the green curve remaining consistently above the baseline. This superiority is particularly pronounced for the most challenging subjects (e.g., Subjects 8, 49, and 7); while the baseline model’s performance degrades towards 65%, the proposed framework exhibits a corrective behavior, stabilizing accuracies above 71% achieving the highest accuracy with ∼78% for subject 7. Furthermore, the narrower confidence intervals (shaded regions) observed for EEG-GCIRNet indicate that the variational regularization effectively mitigates the variability inherent in complex 5-class decision boundaries, yielding not only higher accuracy but greater predictive stability than standard temporal decoding.

Overall, these results confirm that the proposed unimodal VAE framework is not limited to simple binary discrimination but effectively scales to multi-class scenarios, leveraging the latent disentanglement of connectivity patterns to resolve complex motor imagery tasks.

For the multi-class setting, we analyzed the stability of the models through average rankings and assessed the significance of the performance differences using pairwise Wilcoxon signed-rank tests. The summary of these statistical metrics is presented in Table 6. EEG-GCIRNet achieves a perfect average ranking of 1.00, indicating that it consistently outperformed all reference architectures across the considered subjects. This places it significantly ahead of the nearest competitor, TCFusionNet, which obtained an average ranking of 2.15, and well beyond the standard end-to-end baselines like DeepConvNet (3.10) and EEGNet (4.30). The statistical reliability of these gains is confirmed by the hypothesis testing analysis. The proposed model yields the lowest average *p*-value (0.002), which is markedly below the standard significance threshold (α=0.05). While TCFusionNet and DeepConvNet also exhibit statistical significance (p=0.021 and p=0.026, respectively), the order-of-magnitude difference in favor of EEG-GCIRNet underscores its robustness. Conversely, shallow and compact architectures such as ShallowConvNet and EEGNet show high *p*-values (0.25), suggesting that their performance fluctuations in this complex 5-class scenario lack statistical consistency compared to the proposed approach.

### 4.4. Interpretability and Internal Model Dynamics

To understand the mechanisms behind EEG-GCIRNet’s robust performance, we analyzed its internal dynamics and learned representations. The results reveal that the model is not a “black box” but a well-designed framework that adapts its learning strategy, captures physiologically meaningful patterns, and creates a highly effective feature space.

#### 4.4.1. Adaptive Learning Through Loss Weight Reorganization

A key feature of EEG-GCIRNet is its ability to adapt its optimization priorities based on signal quality. The distributions of the three loss component weights, presented in Figure 9, reveal a distinct internal reorganization between the “Good” and “Mid” performance groups for the bi-class scenario. This behavior highlights the model’s inherent ability to promote a balanced interaction between reconstruction accuracy, discriminative capacity, and latent space stability. For the “Good” group, the weights for reconstruction (REC), classification (CLA), and latent space regularization (REG) are relatively balanced (modes: 0.3433, 0.3390, and 0.3824, respectively). The slight predominance of the REG component suggests a focus on maintaining a coherent latent structure, which aligns with recent findings on the importance of regularization for robust performance [59].

In contrast, for the “Mid” group, the model distinctly reorganizes its priorities. The REG component remains dominant (mode: 0.3732), but the CLA weight is significantly reduced (mode: 0.2382) in favor of REC (mode: 0.3519). This numerical shift, visible as a change in the central tendency of the distributions in Figure 9, indicates that when faced with less discriminative signals, the model prioritizes learning a stable and faithful representation of the input data over immediate classification accuracy. This adaptive strategy, where internal representation optimization substitutes for explicit discriminative signals, is a known characteristic of robust unimodal systems [49].

Likewise, we analyzed the learned loss component weights ($\lambda_{\mathrm{REC}}$, $\lambda_{\mathrm{CLA}}$, $\lambda_{\mathrm{REG}}$) for the multi-class scenario, as detailed in Table 7. The distribution of these weights corroborates the adaptive optimization strategy observed in the binary task, but reveals distinct behaviors necessitated by the increased complexity of the 5-class problem. For the highest-performing subject (Subject 13), the model converges to a purely discriminative state ($\lambda_{\mathrm{CLA}}=1.0000$, $\lambda_{\mathrm{REC}}=\lambda_{\mathrm{REG}}=0$), indicating that the spatio-spectral features were sufficiently distinct to drive classification without the need for auxiliary regularization. Conversely, for subjects with moderate performance (e.g., S50, S12, S40), the model adopts a hybrid strategy, balancing classification importance ($\lambda_{\mathrm{CLA}}\approx 0.60$) with reconstruction fidelity ($\lambda_{\mathrm{REC}}\approx 0.40$), while keeping the latent regularization term deactivated ($\lambda_{\mathrm{REG}}=0$). This suggests that for reasonably separable data, the autoencoder acts primarily as a feature extractor rather than a generative regularizer.

However, a critical shift occurs for the most challenging subjects (e.g., 7, 3, 8), where the baseline methods struggled. Here, the model explicitly activates the latent space regularization ($\lambda_{\mathrm{REG}}>0.09$), reaching values up to 0.1753 for Subject 8. This indicates that when decision boundaries are ambiguous, EEG-GCIRNet automatically prioritizes the formation of a well-structured Gaussian latent space to prevent overfitting. An extreme case is observed for Subject 49, where the model shifts almost entirely to representation learning ($\lambda_{\mathrm{REC}}=0.8303$), effectively operating as a non-linear denoiser to stabilize the input features before attempting classification. This evident reorganization of optimization priorities confirms that the architecture is not a static “black box,” but an adaptive system that modulates its learning objective based on the difficulty of the underlying neural patterns.

#### 4.4.2. Qualitative Analysis of Learned Representations and Functional Connectivity

The adaptive weighting strategy of the proposed loss function directly influences the quality of the learned representations, which can be assessed by analyzing the decoder’s output. Figure 10 and Figure 11 present the class-specific topographic reconstructions for a representative “Good” subject (Subject 14) and a “Mid” subject (Subject 27). For Subject 14, the reconstructions are spatially homogeneous, reflecting the “Good” group’s balanced optimization where the model maintains a stable, regularized latent space. Conversely, for Subject 27, the reconstructions appear sharper and more structurally defined. This aligns perfectly with the “Mid” group’s adaptive increase in the reconstruction weight ($\lambda_{\mathrm{REC}}$), which forces the model to prioritize structural fidelity to capture the more complex or noisy signal distributions inherent to this group. The ability to generate coherent and class-consistent topographic maps demonstrates that the VAE successfully captures relevant spatio-spectral patterns.

To validate that these reconstructed topographies represent genuine neural interactions rather than artifacts, we analyzed the underlying functional connectivity networks. Figure 12 illustrates the average connectivity patterns for the same representative subjects (14 and 27), organized by anatomical region and hemisphere. The edges represent the strongest functional connections (95-th to 100-th percentile), revealing distinct network topologies associated with performance levels.

In the case of the high-performing Subject 14, the connectivity patterns reveal a highly integrated network. In the μ band, relevant connections link frontal regions to the right central area, and from there to posterior regions (e.g., P2 to Pz), suggesting a cohesive fronto-centro-parietal network. As the frequency increases into the β bands ($\beta_{l}$, $\beta_{m}$, $\beta_{h}$), this integration intensifies. We observe consistent inter-hemispheric links among posterior regions and salient connections bridging midline and left frontal regions (e.g., Fz–F1). This coordinated involvement of bilateral frontal cortices and the sustained posterior coupling supports the notion that high performance is driven by robust long-range synchronization.

In contrast, the connectivity profile of the mid-performing Subject 27 indicates a different neural strategy, necessitating the model’s enhanced focus on reconstruction. Across the μ and low-β bands, connections are heavily concentrated along the central and centro-parietal midline (particularly C1–Cz and CP1–CPz), with a prominent additional link emerging between Pz and P1. Unlike the broad integration seen in the good performer, this subject exhibits a redistribution of connectivity toward fronto-central and centro-posterior networks. In the mid- and high-β bands, this configuration remains stable, highlighting a sustained involvement of left posterior parietal regions.

The persistence of these specific frequency-dependent connections over sensorimotor and parietal areas in both subjects confirms that the reconstructed maps are biologically valid. The model effectively captures the integrated fronto-parietal networks typical of high performers, as well as the more localized, midline-focused compensatory networks of mid performers. This demonstrates that the EEG-GCIRNet does not simply memorize input images, but learns to encode and reconstruct the underlying physiological connectivity drivers of motor imagery.

#### 4.4.3. Structure and Separability of the Latent Space

The ultimate outcome of the model’s adaptive learning and representation quality is a well-structured and discriminative latent space. To visualize this, t-SNE projections were applied to the latent representations of subjects 14 and 27 (Figure 13).

For the high-performing subject 14 (Figure 13a), the projection reveals two clearly differentiated and cohesive clusters corresponding to the left- and right-hand MI classes. This well-defined separation confirms that the model has learned a highly discriminative latent space, which is consistent with the subject’s high classification accuracy. For the mid-performing subject 27 (Figure 13b), the classes are still largely separable, though with some regions of overlap. This is typical for signals with lower quality, where features are partially discriminative but affected by noise [47,48,60].

Taken together, these visualizations confirm that EEG-GCIRNet’s VAE-based design successfully creates a more structured and discriminative latent space than is typically achievable with standard architectures. This underscores the critical role of latent space regularization in adapting to inter-subject variability and improving generalization, even in the presence of noisy inputs [61,62].

#### 4.4.4. Layer-Wise Relevance Analysis via Grad-CAM++

Figure 14 displays the Grad-CAM++ relevance maps in the channel–subject plane for each of the three convolutional layers of the proposed architecture (conv1, conv2, and conv3) and for both MI classes. Unlike aggregate measures, this visualization presents the CAM corresponding to the model trained on that specific subject separately for each layer and class. It is important to note that these maps do not represent static filter weights; rather, they quantify the spatial contribution of each electrode to the model’s decision, computed via the gradient-weighted combination of feature maps.

In the first convolutional layer (conv1), both classes exhibit broadly distributed activations across subjects and channels, indicating that this layer primarily captures generic spatio-spectral patterns. As the data propagates to conv2, the contribution becomes noticeably more focal, exhibiting clearly delineated channel-wise regions that depend on the MI class. This shift suggests that the model begins to combine spatial and channel information in a more specific manner. Finally, in the deepest layer (conv3), the CAMs become highly sparse and localized, concentrating on a small subset of subjects and channels. This behavior is consistent with a higher degree of specialization, where the deepest layer extracts highly discriminative features directly linked to the final MI decision.

Importantly, the CAM amplitudes are not trivially explained by the classification accuracy of each subject. Some subjects with high MI performance still show relatively diffuse or low-amplitude CAMs, whereas others with lower performance may exhibit more pronounced, localized patterns. This indicates that Grad-CAM++ is highlighting how each subject-specific model internally assigns importance to channels to solve the MI task, rather than simply mirroring inter-subject performance differences. Nevertheless, despite training a separate model per subject, the importance maps reveal consistent horizontal bands across subjects, pointing to a degree of cross-subject convergence in the channels that the network considers informative for left- and right-hand MI.

To assess whether the interpretability of the learned features holds in the more complex multi-class scenario, we extended the Grad-CAM++ analysis to the EEGMMIDB dataset. Figure 15 depicts the relevance maps for four representative classes (C1–C4) across the network depth. Consistent with the binary classification findings, the network exhibits a clear hierarchical refinement of spatial contribution. In the initial layer (conv1, Figure 15a–d), the importance distribution is diffuse and spans vertically across nearly all channels for every class. This suggests that the shallow layers are primarily responsible for extracting low-level, generic spectral features shared among all motor tasks. However, as information flows to the deeper layers (conv3, Figure 15i–l), a high degree of spatial specialization emerges.

Notably, the multi-class scenario requires the model to resolve finer topological differences than simple hemispheric lateralization. The maps in the deepest layer (conv3) demonstrate that the model successfully learns distinct spatial signatures for each class. For instance, the channels contributing to Class 1 (Figure 15i) form a specific cluster that differs topographically from the channels driving the decision for Class 3 (Figure 15k). This spatial disentanglement confirms that EEG-GCIRNet solves the complex 5-class problem by isolating specific subsets of task-relevant electrodes—likely corresponding to the distinct cortical representations of hands and feet—rather than relying on global signal artifacts.

## 5. Discussion

The results obtained with the proposed EEG-GCIRNet architecture provide valuable insights into how model design and latent regularization can jointly address key challenges in motor imagery decoding. This study demonstrated that by transforming functional connectivity into an image-based representation and processing it with a variational autoencoder, it is possible to create a BCI framework that is not only highly accurate but also robust and interpretable. The model achieved remarkable performance across subjects, reaching the highest average accuracy (81.82%) and the lowest inter-subject variability among all evaluated methods, confirming the efficacy of this unimodal, VAE-based approach.

A key contribution of this work lies in its direct response to the critical challenges of inter-subject variability and “BCI illiteracy”. The most compelling finding was the complete elimination of the “Bad” performance group (Figure 6c), coupled with substantial accuracy gains of 21.55% and 13.70% for the “Bad” and “Mid” groups, respectively (Table 4). This demonstrates the profound robustness of EEG-GCIRNet in handling noisy or low-separability EEG signals, where conventional architectures typically fail. By elevating the performance of these challenging subjects, the framework serves a corrective function, suggesting a promising path toward more inclusive and reliable BCI systems that can adapt to a wider range of users. Furthermore, the scalability of the proposed framework is evidenced by its superior performance in the complex 5-class EEGMMIDB scenario, where it achieved an average accuracy of 75.20%, surpassing advanced temporal-channel fusion models by approximately 6.3%. Unlike binary tasks, where hemispheric lateralization is often sufficient for discrimination, the 5-class problem requires the model to resolve finer topological differences between spatially overlapping classes, such as “Both Hands” versus single-hand imagery.

The mechanism underlying this robustness appears to be the model’s sophisticated, adaptive learning strategy. The analysis of the loss weight distributions (Figure 9) revealed that EEG-GCIRNet is not a static “black box” but dynamically reorganizes its optimization priorities based on signal quality. In the bi-class scenario, for subjects with high-quality signals, it maintains a harmonious balance between reconstruction, classification, and regularization. For subjects with more challenging signals, it strategically prioritizes representation learning (REC loss) over immediate classification (CLA loss). This intelligent trade-off is visually confirmed by the qualitative analysis of the reconstructions (Figure 10 and Figure 11), which shows that the model consistently generates spatially coherent and physiologically plausible connectivity maps. This behavior supports the notion that a well-regularized and accurately reconstructed representation is a prerequisite for effective classification, especially in noisy conditions. Moreover, as observed in Table 7, for the most challenging subjects (e.g., Subject 49) in multi-class scenario, the network implicitly shifts its role from a classifier to a non-linear denoiser ($\lambda_{\mathrm{REC}}\approx 0.83$), prioritizing the reconstruction of a clean latent representation before attempting to separate complex decision boundaries. This suggests that the connectivity-driven topographic maps provide a sufficiently rich feature space to support multi-class decoding, provided that the training objective is dynamically regularized to handle the increased cognitive load and signal ambiguity.

The above is further corroborated by the functional connectivity analysis, which reveals distinct network topologies associated with performance levels (see Figure 12). For high-performing subjects, the observed integrated fronto–centro–parietal network supports established findings on the critical role of fronto–parietal and parieto–occipital connectivity in supporting MI performance [63], while the specific beta-band synchronization over central and parietal regions aligns with literature identifying these features as informative markers for task discrimination [64]. Conversely, the connectivity profile of mid-performing subjects suggests a redistribution toward fronto–central and centro–posterior networks; notably, the persistence of μ/β-band connections over sensorimotor and parietal areas in these subjects remains consistent with studies highlighting the fundamental role of these networks in effective BCI control [65,66].

The ultimate outcome of this adaptive process is the creation of a well-structured and discriminative latent space. The t-SNE visualizations (Figure 13) provide direct evidence that the encoder successfully learns to disentangle the features of different MI classes, creating clearly separated clusters for high-performing subjects and maintaining reasonable separation even for mid-performing subjects. This demonstrates that the variational formulation and structured latent regularization are the primary drivers of the model’s superior performance, allowing it to move beyond the limitations of conventional architectures that often struggle with noisy or overlapping feature distributions.

To verify that these learned latent structures correspond to genuine neural mechanisms rather than artifacts, we examined the spatial contribution patterns identified by the network. The layer-wise analysis of the relevance maps reveals consistent spatial structures that align with anatomically meaningful channel groups (see Figure 14 and Figure 15). High-contribution areas in the superior frontal region likely reflect higher-order cognitive control and attentional processes facilitating MI, consistent with reports of prefrontal involvement linked to imagery quality [67,68,69]. Meanwhile, prominent bands of importance spanning fronto-central and parietal sites correspond to the well-known fronto-parietal MI network—encompassing premotor and parietal regions recruited during kinesthetic and visual MI [70] and align with established evidence of alpha and beta modulation and information flow across these regions [71]. Thus, the specific emphasis on parieto-occipital electrodes in the deepest layers suggests the network leverages visuo-spatial components alongside sensorimotor rhythms, a view supported by studies indicating occipital recruitment and connectivity during visual-motor imagery [72]. Regarding the multi-class scenario, the extended Grad-CAM++ analysis confirms that EEG-GCIRNet addresses this complexity through hierarchical spatial refinement, evolving from generic spectral features in shallow layers to highly specialized, class-specific electrode clusters in deep layers. This capability is underpinned by the model’s adaptive optimization strategy, which becomes even more pronounced in this high-complexity setting. Overall, these results confirm that the proposed architecture converges towards neurophysiologically plausible channel patterns, emphasizing networks repeatedly implicated in motor simulation.

Finally, the statistical analysis confirms the performance advantage of EEG-GCIRNet across different levels of task complexity. While the model secured a leading average ranking of 2.32 (p=0.007) in the binary scenario (Table 3), its dominance became even more pronounced in the multi-class evaluation (Table 6). In this challenging 5-class setting, EEG-GCIRNet achieved a perfect average ranking of 1.00 and a highly significant average Wilcoxon *p*-value of 0.002, establishing a statistically clear superiority over advanced architectures like TCFusionNet and DeepConvNet. This indicates that the observed performance gains are not random fluctuations but a consistent outcome of the model’s design. By learning robust latent structures directly from connectivity-based topographic representations, EEG-GCIRNet emerges as a reliable, interpretable, and computationally efficient framework for motor imagery decoding. It effectively balances accuracy, generalization, and representational stability, allowing it to adapt to diverse subject profiles and maintain strong performance across varying signal quality conditions while preserving a physiologically consistent representational organization.

## 6. Concluding Remarks

In this work, we introduced EEG-GCIRNet, a novel unimodal framework based on a variational autoencoder designed to process topographic representations of functional connectivity for motor imagery classification. Through a comprehensive analysis, we demonstrated that this approach effectively addresses the persistent challenges of low spatial resolution, susceptibility to noise, and high inter-subject variability in EEG-based BCIs. By integrating Gaussian functional connectivity with a generative VAE architecture, our method creates a rich, spatially structured representation that allows for simultaneous classification and reconstruction. The findings confirm that our proposed method not only sets a new benchmark for performance but also provides a robust and transparent solution grounded in neurophysiology.

The primary contribution of this study is the demonstration of superior classification performance and the effective mitigation of “BCI illiteracy”. EEG-GCIRNet achieved the highest average accuracy (81.82%) and, critically, the lowest inter-subject variability (±10.15) among all evaluated state-of-the-art models in the binary GigaScience benchmark. Perhaps the most significant finding is the model’s corrective capability: it completely eliminated the “Bad” performance group (subjects with <60% accuracy) and provided substantial gains of ∼22% for these challenging users. This result demonstrates notable statistical significance (p=0.007) and suggests that the framework’s variational nature allows it to extract discriminative information even from noisy or low-separability signals. Furthermore, the framework exhibited remarkable scalability in the complex 5-class EEGMMIDB scenario, achieving a perfect average ranking of 1.00 and a highly significant Wilcoxon *p*-value of 0.002 against advanced baselines. This confirms that EEG-GCIRNet maintains its robustness even when resolving finer topological differences between spatially overlapping classes.

This robust performance is driven by the model’s sophisticated, adaptive learning strategy. The analysis of loss component weights revealed that EEG-GCIRNet is not a static “black box” but dynamically reorganizes its optimization priorities based on the quality of the input signal. For high-performing subjects, the model maintains a harmonious balance between reconstruction, classification, and regularization. However, for subjects with weaker signals (the “Mid” group), or in high-complexity multi-class tasks, the model automatically shifts its focus, prioritizing representation learning (reconstruction loss) over immediate classification accuracy. This intelligent trade-off allows the model to act as a non-linear denoiser, ensuring a stable feature encoding before attempting classification. The ultimate outcome of this process is a well-structured latent space where MI classes are clearly disentangled, as evidenced by the distinct clustering observed in the t-SNE projections.

Furthermore, our extended interpretability analysis provides direct physiological validation of the learned representations, confirming that the model’s decisions are based on genuine neural mechanisms. The layer-wise Grad-CAM++ visualization demonstrated a hierarchical learning process: the model progresses from capturing generic spectral features in shallow layers to extracting highly localized, discriminative patterns in deep layers, specifically targeting sensorimotor and parieto-occipital regions. Crucially, the functional connectivity analysis confirmed that these patterns are biologically meaningful. The model correctly identifies the integrated fronto–centro–parietal networks, characterized by long-range synchronization, which are typical of high-performing subjects. Conversely, for mid-performing subjects, it detects distinct compensatory mechanisms, characterized by localized, midline-focused connectivity patterns. This confirms that EEG-GCIRNet leverages genuine neurophysiological mechanisms—such as μ/β rhythm modulation and network integration—rather than relying on artifacts or spurious correlations.

Despite the promising results, this study has several limitations that must be acknowledged. First, while the framework was validated on two distinct datasets (GigaScience and EEGMMIDB), the evaluation was restricted to motor imagery paradigms. This constrains the generalizability of our findings to other BCI paradigms (e.g., P300, SSVEP) with different temporal characteristics. Second, although the model demonstrated adaptive behavior, the hyperparameters weighting the loss components were static. This fixed weighting may not be optimal for all subjects, and no strategies for dynamic, performance-based adaptation were explored. Third, our framework’s reliance on topographic map generation means that it explicitly requires electrode position coordinates as an input. This requirement for spatial metadata is not shared by some benchmark models, such as those that process the EEG signal as a simple (channels × time) matrix. While this could be considered a limitation in a hypothetical scenario where such standard information is unavailable, we frame it as a deliberate methodological choice. Finally, our interpretability analysis, while insightful, remains indirect. The analysis of reconstructions and latent space provides a high-level understanding but does not offer the granular, feature-level attribution that techniques like attention-based visualization can provide.

The findings and limitations of this study open several avenues for future research. To address generalizability, the next logical step is to evaluate the EEG-GCIRNet framework on larger-scale benchmarks such as the BCI Competition IV-2a. This will allow us to assess the model’s performance and the clarity of its VAE-based visualizations when handling more complex decision boundaries on widely standardized data. Furthermore, we propose exploring extensions of the model that incorporate dynamic weighting mechanisms for the loss components, which could allow for even greater subject-specific adaptation. A further step will be to statistically validate the model’s internal dynamics; while our analysis of the loss component weights reveals a clear reorganization of priorities, future work will include formal statistical testing to quantify the significance of these shifts between performance groups. We also plan to investigate hybrid architectures, particularly those based on Transformers, to enhance the fusion of raw temporal EEG signals with our connectivity-derived topographic maps, potentially creating a more powerful, end-to-end model. Finally, integrating complementary interpretability techniques, such as feature attribution or attention-based visualization methods, will be crucial for providing a more precise understanding of the model’s decision-making process at the clinical level.

## Figures and Tables

**Figure 1 sensors-26-00227-f001:**
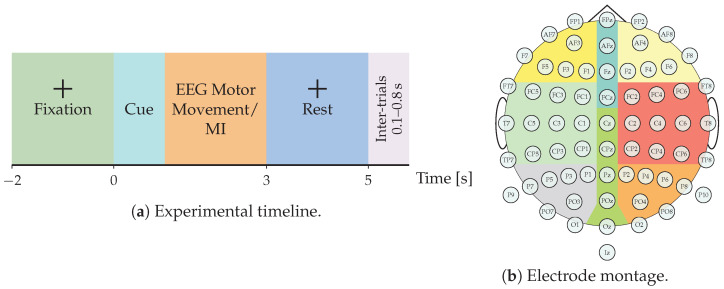
Overview of the GigaScience MI-EEG dataset: experimental timeline (**a**) and electrode configuration (**b**) where colors indicate scalp regions: 
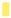
 Frontal left; 
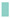
 Frontal; 
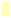
 Frontal right; 
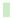
 Central left; 
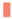
 Central right; 
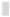
 Posterior left; 
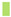
 Posterior; 
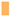
 Posterior right.

**Figure 2 sensors-26-00227-f002:**
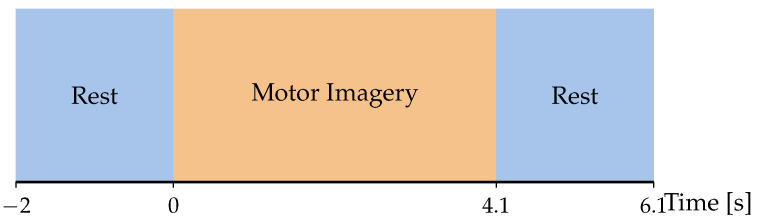
Overview of the EEGMMIDB MI-EEG dataset: experimental timeline. The electrode montage shares the same 64-channel configuration as the GigaScience dataset in Figure 1.

**Figure 3 sensors-26-00227-f003:**
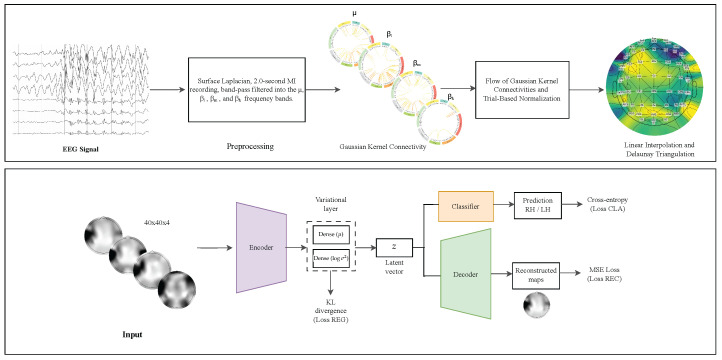
The proposed EEG-GCIRNet framework, composed of two main stages: (**Top**): A feature engineering pipeline that transforms raw EEG signals into multi-channel topographic maps using GFC. (**Bottom**): A VAE architecture that processes these maps to jointly learn reconstruction, classification, and regularization from a shared latent space.

**Figure 4 sensors-26-00227-f004:**
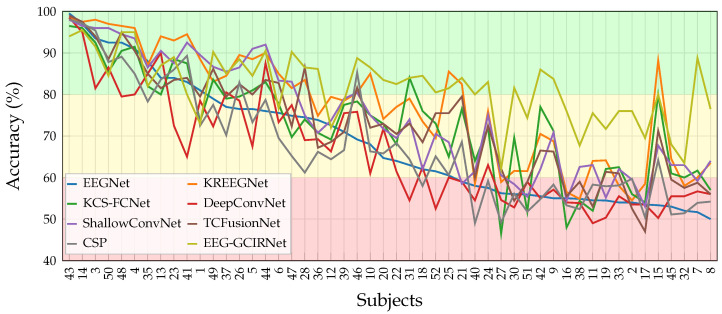
Inter-subject accuracy results. Subjects are sorted based on EEGNet performance.The background colors indicate subject performance groups: red corresponds to the “Bad” group (accuracy ≤ 60%), yellow to the “Mid” group (60% < accuracy ≤ 80%), and green to the “Good” group (accuracy > 80%).

**Figure 5 sensors-26-00227-f005:**
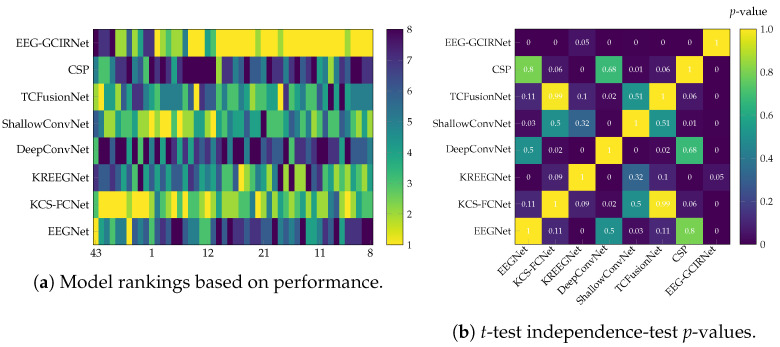
Models rankings vs. *t*-test *p*-values. In (**a**) subjects are sorted based on EEGNet’s accuracy. In (**b**) the matrix of *p*-values from post-hoc pairwise t-tests are corrected for multiple comparisons using the Holm–Bonferroni method.

**Figure 6 sensors-26-00227-f006:**
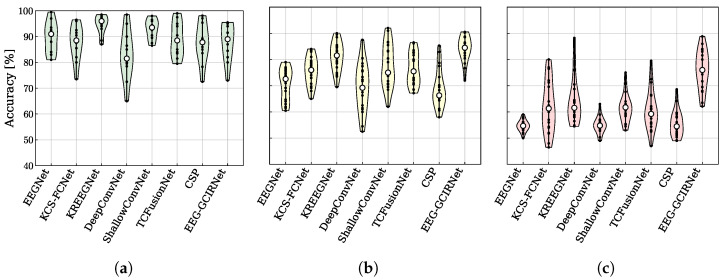
Group performing MI-EEG classification results. The white dot in each violin plot indicates the median accuracy, while the thick black bar represents the interquartile range. The corresponding average (mean) accuracies for EEGNet and EEG-GCIRNet are reported in Table 4. (**a**) Subjects with EEGNet accuracy above 80%. (**b**) Subjects with EEGNet accuracy between 60% and 80%. (**c**) Subjects with EEGNet accuracy below 60%.

**Figure 7 sensors-26-00227-f007:**
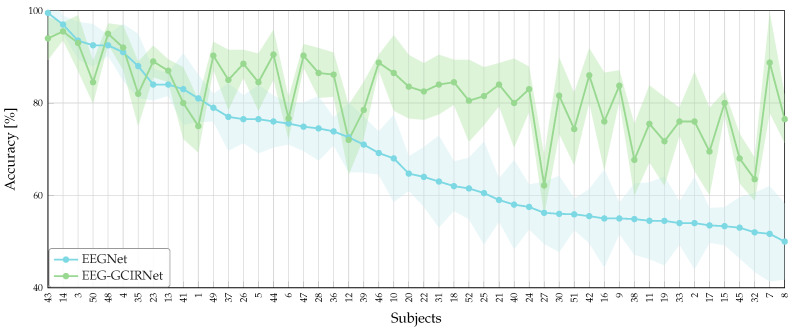
Inter-subject classification accuracies of EEGNet and EEG-GCIRNet. Each point represents the average performance per subject, with subjects ordered according to EEGNet accuracy to highlight the comparative improvements achieved by EEG-GCIRNet. Shaded regions indicate performance variability.

**Figure 8 sensors-26-00227-f008:**
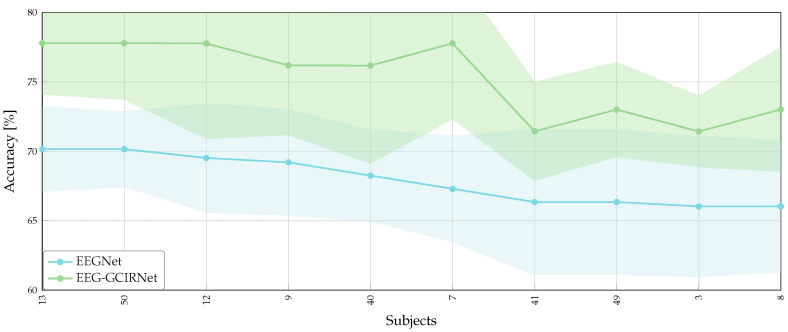
Inter-subject classification accuracies for EEGNet and EEG-GCIRNet in the 5-class MI-EEG setting. Subjects are ordered according to their EEGNet accuracy to emphasize the consistent performance gains achieved by EEG-GCIRNet. Shaded regions denote the accuracy variability across trials.

**Figure 9 sensors-26-00227-f009:**
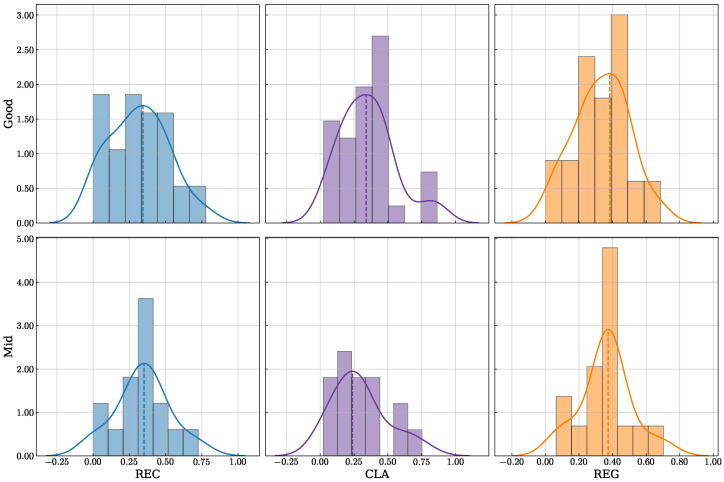
Distributions of the weights corresponding to the three loss components—reconstruction (REC), classification (CLA), and latent space regularization (REG)—for the performance groups “Good” and “Mid”. Solid curves represent the density estimates, while dashed vertical lines indicate the mode of each distribution.

**Figure 10 sensors-26-00227-f010:**
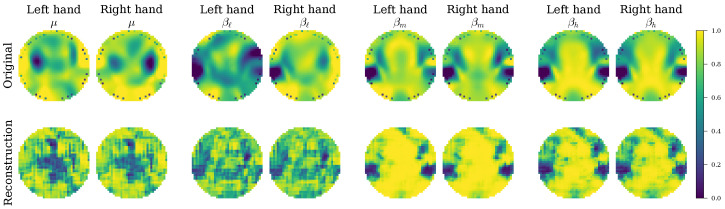
Reconstruction subject 14 corresponding to group “Good”.

**Figure 11 sensors-26-00227-f011:**
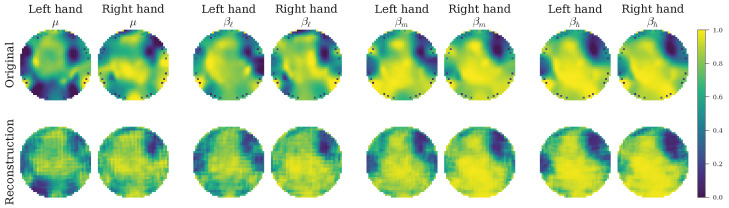
Reconstruction subject 27 representing group “Mid”.

**Figure 12 sensors-26-00227-f012:**
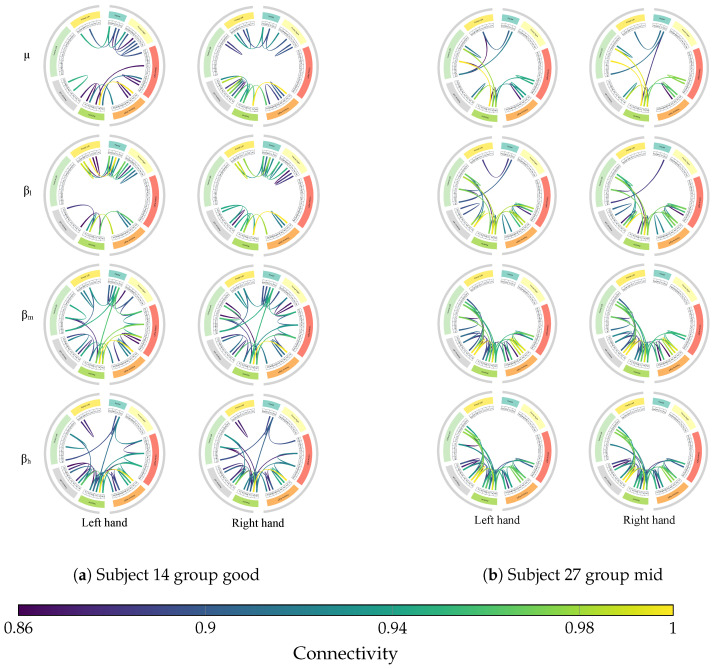
Functional connectivity patterns for two representative subjects. For each subject, the left and right columns correspond to left- and right-hand motor imagery, respectively, and the rows correspond to the μ, $\beta_{l}$, $\beta_{m}$, and $\beta_{h}$ bands.

**Figure 13 sensors-26-00227-f013:**
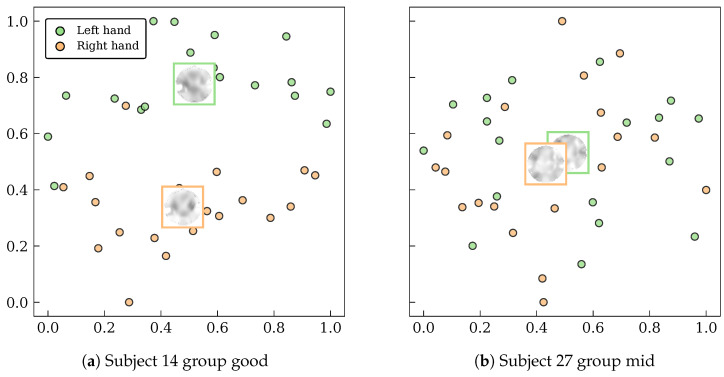
t-SNE of latent representations by performance group, good and mid. Colors correspond to the motor imagery classes (left hand and right hand).

**Figure 14 sensors-26-00227-f014:**
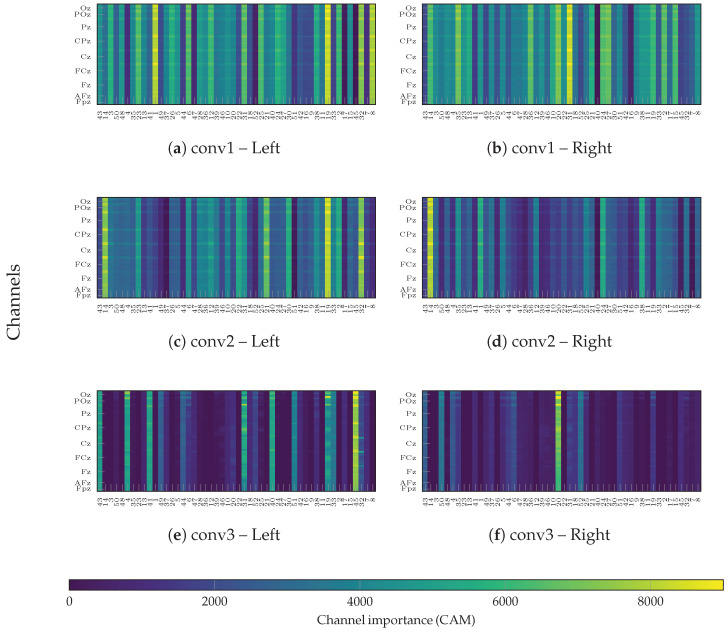
Channel-wise CAM contribution across subjects for each convolutional layer and MI class. The color intensity represents the relative importance of each channel in the model’s decision-making process.

**Figure 15 sensors-26-00227-f015:**
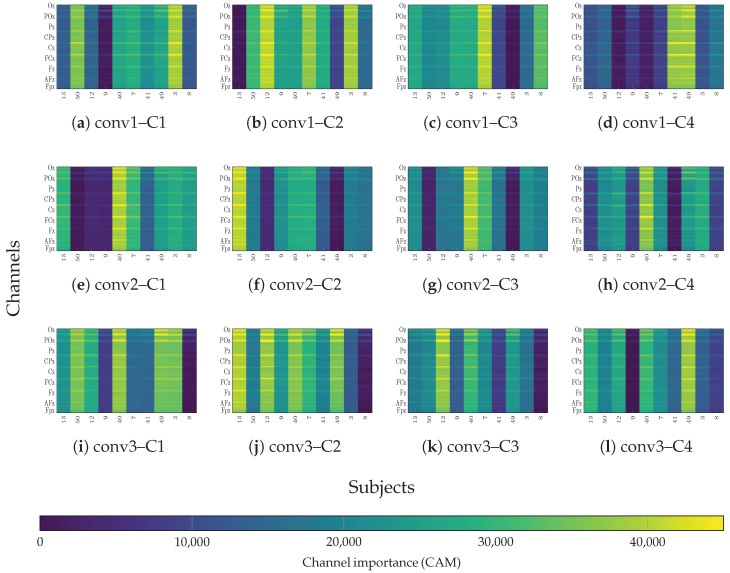
Channel-wise class activation maps (CAMs) for motor imagery (MI) classes across convolutional layers. Rows correspond to *conv1*, *conv2*, and *conv3*, while columns (C1–C4) denote the selected MI classes. Color intensity reflects the relative contribution of each EEG channel, using a shared global color scale.

**Table 1 sensors-26-00227-t001:** Layer-wise configuration of the EEG-GCIRNet architecture. The input shape corresponds to the four-channel topographic maps ($\tilde{H}\times \tilde{W}\times C$).

Block	Layer	Kernel/Units	Strides	Activation	Output Shape
**Input Image**					
					(40,40,4)
**Encoder**	Conv2D	3×3, 6 Filters	(1,1)	SELU	(40,40,6)
($E_{\phi }$)	AvgPool2D	2×2	(2,2)	-	(20,20,6)
	Conv2D	3×3, 16 Filters	(1,1)	SELU	(20,20,16)
	AvgPool2D	2×2	(2,2)	-	(10,10,16)
	Conv2D	3×3, 120 Filters	(1,1)	SELU	(10,10,120)
	Flatten	-	-	-	(12,000)
	Dense	128 Units	-	SELU	(128)
**Latent Space**	Dense (μ)	128 Units	-	Linear	(128)
(Reparameterization)	Dense ($\log\sigma^{2}$)	128 Units	-	Linear	(128)
**Decoder**	Dense	128 Units	-	SELU	(128)
($D_{\theta }$)	Dense	12,000 Units	-	SELU	(12,000)
	Reshape	(10,10,120)	-	-	(10,10,120)
	Conv2DTranspose	3×3, 16 Filters	(1,1)	SELU	(10,10,16)
	Upsampling	2×2	(2,2)	-	(20,20,16)
	Conv2DTranspose	3×3, 6 Filters	(1,1)	SELU	(20,20,6)
	Upsampling	2×2	(2,2)	-	(40,40,6)
	Reconstruction	3×3	-	Sigmoid	(40,40,4)
**Classifier**	Dense	128 Units	-	SELU	(128)
($C_{\psi }$)	Dense (Output)	2 Units	-	Softmax	(2)

**Table 2 sensors-26-00227-t002:** MI-EEG classification performance comparison: Average ACC ± standard deviation. The standard deviation represents the variability of classification accuracy across the 50 subjects.

Model	ACC [%]
CSP [20]	67.66±13.81
EEGNet [52]	68.39±15.50
KREEGNet [27]	77.32±14.74
KCS-FCNet [38]	72.77±13.53
DeepConvNet [53]	66.55±14.24
ShallowConvNet [54]	74.56±14.60
TCFusionNet [55]	72.81±14.10
EEG-GCIRNet (Ours)	81.82±10.15

**Table 3 sensors-26-00227-t003:** Average rankings and *p*-values.

Model	Avg. Ranking	Avg. T-Test *p*-Value
CSP	6.30	0.23
EEGNet	5.76	0.22
KCS-FCNet	4.58	0.25
KREEGNet	2.48	0.07
DeepConvNet	6.42	0.17
ShallowConvNet	3.40	0.19
TCFusionNet	4.34	0.26
EEG-GCIRNet (Ours)	2.32	0.007

**Table 4 sensors-26-00227-t004:** Accuracy and Gain by Group and Approach.

Approach	Group	Accuracy (%)	Gain (%)
	Good	89.64	–
EEGNet	Mid	70.54	–
	Bad	54.65	–
	Good	87.86	−1.78
EEG-GCIRNet (Ours)	Mid	84.24	13.70
	Bad	76.20	21.55

**Table 5 sensors-26-00227-t005:** MI-EEG 5-class classification performance comparison: Average ACC ± standard deviation across the 10 considered subjects.

Model	ACC [%]
EEGNet [52]	67.94±3.98
DeepConvNet [53]	68.35±4.11
ShallowConvNet [54]	67.90±4.06
TCFusionNet [55]	68.89±3.90
EEG-GCIRNet (Ours)	75.20±4.63

**Table 6 sensors-26-00227-t006:** Average rankings and average pairwise Wilcoxon *p*-values.

Model	Avg. Ranking	Avg. Wilcoxon *p*-Value
EEGNet	4.30	0.25
DeepConvNet	3.10	0.026
ShallowConvNet	4.45	0.25
TCFusion	2.15	0.021
EEG-GCIRNet (Ours)	1.00	0.002

**Table 7 sensors-26-00227-t007:** Subject-specific weight distribution for reconstruction (REC), classification (CLA), and latent regularization (REG) across the selected subjects of the EEGMMIDB dataset.

Subject	REC	CLA	REG
13	0.0000	1.0000	0.0000
50	0.5000	0.5000	0.0000
12	0.3655	0.6345	0.0000
9	0.3803	0.6197	0.0000
40	0.3559	0.6441	0.0000
7	0.4958	0.3824	0.1219
41	0.5401	0.4583	0.0017
49	0.8303	0.1697	0.0000
3	0.4421	0.4615	0.0964
8	0.3869	0.4379	0.1753

## Data Availability

The databases used in this study are public and can be found at the following links: http://gigadb.org/dataset/100295 (accessed on 1 July 2025) and https://physionet.org/content/eegmmidb/1.0.0/ (accessed on 7 December 2025).

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
