# Peer review of "Gaussian Connectivity-Driven EEG Imaging for Deep Learning-Based Motor Imagery Classification"

_sensors, 2025, doi:10.3390/s26010227_

Round 1
Reviewer 1 Report
Comments and Suggestions for Authors
The study proposes a Gaussian connectivity-driven EEG imaging representation network, which utilizes a variation autoencoder framework based on the LeNet architecture to improve motor imagery classification accuracy and to provide the interpretability of the proposed framework. The authors validated their approach with a publicly available dataset, which involves EEG data of left and right hand motor imagery. While the paper provides interesting approach, the manuscript lacks sufficient justification and analysis to support their claims. Below are several major comments that could further enhance the manuscript.
Major Comment 1.
A public data from GIGAScience was used to address the efficacy of the proposed model. The dataset focuses on two-class motor imagery, which does not address the model's performance in multi-class classification scenarios. Given that the interpretability is one of the core component of the model, additional analysis using multi-class classifications with accuracy measures and visualizations from VAE seems essential to demonstrate the superiority of the proposed model against others (for example, BCICIV IIa - 4 class motor imagery dataset).
Major Comment 2.
find the rationale for separately analyzing the performance of the models into three distinct groups,“Good,” “Mid,” and “Bad”,to be unclear, particularly on the use of EEGNet's accuracy among the benchmark models. If the intention is to show that the proposed model "maintained high accuracy without overfitting", it may be more effective to use the proposed model itself for the group separation.
- In addition, there seems to be a mismatch between Figure 5 and Table 4, which I understand Table 4 is the numerical results of Figure 5. (Please see EEGNet's accuracy on Figure 5a and the EEGNet's accuracy for "Good" group on Table 4)
Major Comment 3.
The proposed framework involves mutliple hyperparameters (bandwidth σ for the hyperparameter of the gaussian kernel, λ hyperparameters for each of the loss forming the final loss), yet it is unclear how these parameters are determined. It appeared that these parameters were fixed prior to learning, until subsection 4.3.1 indicates that they are adapted through the learning process. Clarifications may be needed, and if the parameters were fixed prior to training, the exact value should be mentioned in the manuscript for reproducibility.
Major Comment 4.
In line with Major Comment 2, I do not see the necessity of comparing the proposed model directly with EEGNet (as shown in Figure 6), and I find this comparison somewhat redundant in relation to Figure 3. The manuscript could be strengthened by utilizing other public datasets to demonstrate the consistency of the model's performance, rather than conducting multiple trainings and retrieving accuracies within each individual subject.
Major Comment 5.
The manuscript states that "The REG component remains dominant, but the CLA weight is significantly reduced". However, I could not find significant differences in the weights between the groups through the manuscript Figure 7. Conducting a statistical analysis and including results for the "Bad" group may provide clarity to support the authors' assertion.
Major Comment 6.
While the paper emphasizes the interpretability of the proposed framework as one of the strengths, the analysis does not move into interpretability in detail, except from representative plots involving data from two subjects. This does not convincingly demonstrate that the VAE successfully captures spatio-spectral patterns. Additional analysis that can clearly address this term along with statistical tests, and quantification of the VAE's performance, and outcomes involving data from all subjects should be included to substantiate this claim.
Minor Comment 1.
It is unclear to me whether the timewindow of the EEG signals were 2.5 seconds or 2 seconds in length, as Figure 2 suggests that 2.5-second MI recordings were used while the description on the second paragraph of subsection 3.1 states that the data is temporally segmented "retaining the window from 2.5 to 4.5 seconds".
Minor Comment 2.
The core component of the proposed model lies in the reconstruction of the topomap alongside the VAE, which not only provides interpretability but also significantly impacts the model's performance. As I understand it, the proposed model (and KREEGNet) utilizes the positions of each electrode to enhance prediction performance, as well as electrode labels for each signal as input. This requirement for additional data information, compared to the other benchmark models, should be acknowledged as a limitation in the discussion section.
Minor Comment 3.
The manuscript conducts multiple comparisons with p-values but does not provide a description of how corrections were applied to these comparisons. This should be addressed.
Minor Comment 4.
In Figure 5 legend, it would be beneficial to state what the black dots and the white dots represent.
Reviewer 2 Report
Comments and Suggestions for Authors
The authors present a model for transforming functional connectivity patterns into a robust image-based representation. These images are used to train a classifier whose design is based on the LeNet architecture, and they also articulate this architecture with a variational autoencoder. Although the document is very comprehensive, in many sections the focus (given by the title) is lost. Likewise, various aspects throughout the manuscript need to be corrected or clarified:
- I think that both the proposed method and the results should focus on the main proposal of the manuscript. Please review and restructure.
- The caption for Figure 2 (Bottom) refers to a VAE. However, Figure 2 (Bottom) does not correspond to the complete VAE. The figure needs to be expanded or supplemented.
- Section 2.5 states that in the encoder, the activation functions of the layers are ReLU, however, Table 1 states that they are SELU. It is necessary to review and correct the inconsistencies in the proposed model.
- Throughout the paper, the number of classes is unclear. Please clarify how many classes there are and what they are.
- Specify what the standard deviation in Table 2 corresponds to (whether it is between classes, between different tests, etc.).
- If the dataset used corresponds to 52 subjects, how was the data distributed? Were results such as those shown in Figure 3 also calculated based on the training data?
Round 2
Reviewer 1 Report
Comments and Suggestions for Authors
The manuscript proposes a Gaussian connectivity-driven EEG imaging representation network for motor imagery classification. While the authors have made some revisions that improved the manuscript, several of my concerns, which I find critical, have not been resolved.
Major 1. Multi-class evaluation: The authors claim their approach is superior to other models. However, this has only been evaluated on a two-class motor imagery problem using a single public dataset. While using only a single dataset is already a considerable limitation especially when claiming that the proposed model outperforms other state-of-the-art models, this single dataset being a two-class problem makes the study shortage of evidence to support the proposed model's superiority.
Major 2. Statistical Analysis for "Significance": P-values must be reported for every instance where a result is claimed to be "significant." Otherwise, alternate term should be used.
Major 3. Grad-CAM++ Analysis: For the Grad-CAM analysis, it would be ideal to address whether the weights of the layers converged toward motor imagery-related channels in order to claim that the weights were learnt properly to "solve the MI task, rather than simply mirroring inter-subject performance differences", as visualization of weight changes across layers do not reflect whether the features were extracted more towards the MI task by using the proposed approach.
I suggest that the manuscript could be published after these major comments are clearly addressed.
